# Factors Associated with Daptomycin-Induced Eosinophilic Pneumonia

**DOI:** 10.3390/antibiotics11020254

**Published:** 2022-02-16

**Authors:** Kazuhiro Ishikawa, Takahiro Matsuo, Yasumasa Tsuda, Mahbubur Rahman, Yuki Uehara, Nobuyoshi Mori

**Affiliations:** 1Department of Infectious Diseases, St. Luke’s International Hospital, Tokyo 104-8560, Japan; takahirom1226@gmail.com (T.M.); yukiue@luke.ac.jp (Y.U.); morinob@luke.ac.jp (N.M.); 2Department of Pharmacy, St. Luke’s International Hospital, Chuo-ku, Tokyo 104-8560, Japan; tsuyasu@luke.ac.jp; 3Division of Epidemiology, Graduate School of Public Health, St. Luke’s International University, Tokyo 104-0045, Japan; rahman@luke.ac.jp; 4Department of Clinical Laboratory, St. Luke’s International Hospital, Tokyo 104-8560, Japan; 5Department of Microbiology, Juntendo University Faculty of Medicine, Tokyo 113-8421, Japan; 6Department of General Medicine, Juntendo University Faculty of Medicine, Tokyo 113-8421, Japan

**Keywords:** daptomycin, eosinophilic pneumonia, *Staphylococcus aureus*, methicillin-resistant *Staphylococcus aureus*

## Abstract

The risk factors for eosinophilic pneumonia (EP) remain unclear. We investigate the characteristics of patients with daptomycin (DAP)-induced EP and conducted a retrospective observational study. A total of 450 patients aged ≥ 18 years who received DAP (25 DAP with EP, 425 DAP without EP) were included. The median duration from the first DAP administration to EP onset was 18.0 days. Definite, probable, and possible DAP-induced EP were diagnosed in 0, 9, and 16 patients, respectively. The median age (DAP with EP, 72.0 years; DAP without EP, 64.0 years), DAP dosage/body weight (BW) (9.00 vs. 7.50 mg/kg), blood eosinophil count (cells/μL) (419 vs. 96), and the percentage of hemodialyzed patients (40.0% vs. 13.4%) were significantly higher in patients with EP than in patients without EP in the univariate analysis. In separate multivariate logistic regression analyses, age (odds ratio (OR), 1.03; 95% confidence interval (CI), 1.00–1.05), DAP dosage/BW (OR, 1.61; 95% CI, 1.25–2.07), and hemodialysis (OR, 4.42; 95% CI, 1.86–10.5) were significantly associated with DAP-induced EP. Clinicians may need to consider the potential factors associated with EP, especially in older patients, patients on hemodialysis, or patients who receive > 9.00 mg/kg of DAP.

## 1. Introduction

Methicillin-resistant *Staphylococcus aureus* (MRSA) infections have high mortality rates of 15–25% [1,2]. The standard treatment for MRSA bacteremia is vancomycin (VCM) or daptomycin (DAP) [3]. DAP is a cyclic lipopeptide antimicrobial agent used to treat skin and soft tissue infections and infective endocarditis. DAP is bactericidal and is considered to be more effective than VCM for MRSA infections; this was observed in a previous study where a higher the duration of negative blood cultures was observed with DAP than VCM [4]. DAP has a high protein-binding concentration of 90–93%, a long half-life in blood (about 8.5 h), and is excreted by the kidneys. For patients with a creatinine clearance (CCr) of ≥ 30 mL/min, the DAP dosage is 4–10 mg/kg in a single daily dose or once every 48 h for patients with a CCr < 30 mL/min or patients on dialysis [5]. The elevation of creatine kinase (CK) levels and eosinophilic pneumonia (EP) are side effects of DAP [5].

EP are characterized by marked accumulations of infiltrating eosinophils in the alveolar space and the interstitium. Although EP can lead to hypoxemia and serious complications, very few studies have been published on its risk factors, including literature reviews and a study with a small sample size [6,7]. These studies reported that older age and a total DAP dose > 10 g are risk factors for DAP-induced EP. Conversely, it has been reported that patients with obesity are more likely to experience adverse effects from beta-lactam antimicrobials due to an increased volumetric distribution (Vd) [8]. However, whether obesity is a risk factor for DAP-induced EP is unknown. In this study, we retrospectively investigate the risk factors for DAP-induced EP.

## 2. Results

A total of 450 patients treated with DAP between June 2011 and July 2020 were selected. Of those, 25 developed EP after the first DAP administration (EP with DAP), while 425 did not (EP without DAP) (Figure 1).

The median duration from the first DAP dose administration to EP onset among the 25 patients with EP was 18.00 (interquartile range (IQR), 26.5; minimum–maximum, (3–49) days. The number of patients with definite, probable, and possible EP is as follows (Table 1).

No patients underwent BAL, so the number of patients with definite EP was zero. Fourteen patients with possible EP improved clinically after the discontinuation of DAP, but two patients died of heart failure.

Table 2 shows the results of the univariate analysis of the baseline and clinical characteristics of both groups.

Compared to the DAP without EP group, the DAP with EP group had the following characteristics, which were statistically significant: a higher median age (IQR) (72.0 (26) vs. 64.0 (32) years, *p* = 0.030), lower median body mass index (BMI) (IQR) (20.0 (6.3) vs. 21.9 (6.0) kg/m^2^, *p* = 0.040), greater proportion of hemodialyzed patients (10 (40%) vs. 57 (13.4%), *p* < 0.001), higher DAP dosage (IQR) (525 (235) vs. 350 (230) mg/day, *p* = 0.046), higher DAP dosage/body weight (IQR) (9.00 (2.7) vs. 7.50 (2.60) mg/kg, *p* < 0.001), higher dosage/ideal body weight (IBW) (IQR) (8.90 (2.70) vs. 7.40 (3.84) mg/kg, *p* = 0.025), higher dosage/adjusted body weight (ABW) (IQR) (8.93 (2.33) vs. 7.44 (3.25) mg/kg, *p* = 0.020), higher total dosage of DAP (IQR) (7875 (9300) vs. 4800 (8100) mg, *p* = 0.032), higher total dosage of DAP (IQR)(7875 (9300) vs. 4800 (8100) mg, *p* = 0.032), higher blood eosinophil count (IQR) (419 (910) vs. 96 (255)/µL, *p* < 0.001), and higher C-reactive protein level (IQR) (10.6 (15.8) vs. 5.39 (9.2) mg/dL, *p* = 0.010). The DAP with EP group had a lower MRSA infection (defined as positive culture of blood, urine, abscess, soft tissue, and so on) proportion, although it was not statistically significant (4 (16.0%) vs. 29 (6.8%), *p* = 0.087). The distribution of DAP dosage/BW (mg/kg) in DAP with EP is shown in Figure 2.

Separate multivariate logistic regression analyses were conducted for each of the characteristics after adjusting for age and BMI (Table 3). Age (odds ratio (OR), 1.03; 95% confidence interval (CI), 1.00–1.05), DAP dosage/BW (OR, 1.61; 95% CI, 1.25–2.07), and hemodialysis (OR, 4.42; 95% CI, 1.86–10.5) were significantly associated with DAP-induced EP.

## 3. Discussion

In this study, we observed that hemodialysis and dosage/BW are associated with EP. The eosinophil count in bronchoalveolar lavage (BAL) is a criterion for the definite diagnosis of EP [6]. However, BAL could not be performed clinically due to the risk of worsening disease progression. Previously published studies used the blood eosinophil count as a criterion for possible EP [6]. In our study, the eosinophil count in blood was significantly higher in the DAP with EP group than in the DAP without EP group. Therefore, we assumed that the blood eosinophil count could be substituted for that of BAL for the diagnosis of EP.

The U.S. FDA identified six cases (patient age range, 60–87 years) of DAP-induced EP between 2004 and 2010 [9,10,11,12]. The patients developed EP 2–4 weeks after DAP initiation, while all seven patients reported improvement or resolution of symptoms after DAP discontinuation. Five of the seven patients were also treated with systemic corticosteroids, and two patients reported EP recurrence after the re-administration of DAP. In another retrospective study [7], the risk factors for DAP-induced EP were age ≥ 70 years and duration of therapy > 14 days. In our study, the median age and the median number of days from DAP initiation to EP onset were identified as risk factors. These results are consistent with those of a previously published report [13]. All the patients except two who died of heart failure improved clinically and rapidly after the treatment with DAP was discontinued. Therefore, none of the patients used steroids for the treatment of EP.

In our study, we found that hemodialysis, DAP dosage/BW, DAP dosage/IBW, DAP dosage/ABW, and total dosage of DAP are significant risk factors for EP. The reported dosage of DAP in a literature review was 4.4–8.0 mg/kg/day [6]. Nevertheless, EP was not observed among the participants of a prospective cohort study of patients with left-sided infective endocarditis treated with DAP at a dose of 9.2 mg/kg/day [4]. In an observational study including 102 patients with infectious endocarditis treated with a DAP dose of 8.2 mg/kg, three patients developed EP [14]. In a retrospective study [7], a total dose of DAP > 10 g was observed as a risk factor in an unadjusted analysis. Hayes et al. hypothesized that DAP may cause EP by binding with the human pulmonary surfactant, resulting in its accumulation in the alveolar spaces at concentrations high enough to injure the epithelium and cause inflammation [10]. DAP has a very small volume of distribution of 7 L, which is indicative of a very little tissue distribution [15]. According to the manufacturer’s instructions, there is no need to adjust the DAP dosage for obese patients. However, when calculating the dosage/BW, it is recommended to use the IBW for hydrophilic drugs and the ABW for lipophilic drugs (mainly for obese people) [16]. In this study, we could not show the optimal dose of DAP for body types, such as obese and lean, but we observed that a high dose is a risk factor for EP. DAP clearance is performed by the kidneys. For patients on dialysis, it is slowly cleared from the body by hemodialysis (approximately 15% of the administered dose is removed in over 4 h) and peritoneal dialysis (approximately 11% of the administered dose is removed in over 48 h). The dosing interval for patients with a creatinine clearance of <30 mL/min should be decreased from every 24 h to every 48 h. However, CK elevation may occur even in patients with renal dysfunction, despite appropriate dosages and intervals [17]. Therefore, EP should be anticipated when administering DAP to patients with renal dysfunction. If possible, other anti-MRSA antibiotics such as vancomycin should be used for hemodialyzed patients.

This study has several limitations. First, this study is based on a small number of events using a single-center observational study design. Therefore, prospective clinical studies are warranted to shed more light on the causal association between identified risk factors and DAP-induced EP. Second, although BAL is a mandatory criterion in the FDA definition of EP, all our patients did not undergo BAL. Since most of the patients had probable or possible EP, the possibility that EP may have had other etiologies than DAP use cannot be completely ruled out. However, blood eosinophil counts were significantly higher in the patients with DAP-induced EP. They improved clinically and rapidly after DAP discontinuation, suggesting that BAL may not be essential for diagnosis

## 4. Materials and Methods

### 4.1. Study Design and Setting

This was a single-center retrospective observational study conducted at St. Luke’s International Hospital, a 520-bed teaching hospital in Tokyo.

### 4.2. Inclusion and Exclusion Criteria

Adult patients (aged ≥ 18 years) who were hospitalized and treated with intravenous DAP between June 2011 and July 2020 were included. Eligible patients were identified from the hospital’s electronic database. In this study, DAP-induced EP was classified as definite, probable, and possible, in accordance with the definition by Kim et al. [6] (Table 4)

The criteria for definite EP are similar to those of EP put forth by the United States Food and Drug Administration (U.S. FDA; all the criteria have to be met): concurrent exposure to DAP, fever, dyspnea with increased oxygen requirement or requiring mechanical ventilation, new infiltrates observed on a chest X-ray or computed tomography scan, BAL with > 25% eosinophils, and clinical improvement following DAP withdrawal.

### 4.3. Data Collection

Patient data were extracted from inpatient electronic medical records. The study variables included patient demographics (age and sex), comorbidities, use of antimicrobials other than DAP, DAP dosage, body weight, height, vital signs, laboratory test results, radiology imaging including chest radiography and computed tomography, microbiological findings, mortality, and complications including septic shock and mechanical intubation.

### 4.4. Statistical Analyses

Bivariate associations were assessed using χ^2^ and Fisher’s exact test for categorical variables and the Mann–Whitney U test for continuous variables. A significance level of 0.10 was set for variables in the univariate analysis to be included in the multivariate analyses. According to the results of the univariate analysis, separate multivariate logistic regression analyses were conducted for each of the characteristics after adjusting for age and BMI. All analyses were performed using SPSS 19.0 J statistical software (IBM Japan, Tokyo, Japan).

### 4.5. Patient Consent Statement

This study was approved by the Institutional Review Board of St. Luke’s International Hospital in Tokyo, Japan (number: 20-R101). The requirement for patient consent was waived due to the study’s retrospective nature.

## 5. Conclusions

Older patients, higher DAP dosage/BW, and hemodialysis were independent risk factors for DAP-induced EP.

## Figures and Tables

**Figure 1 antibiotics-11-00254-f001:**
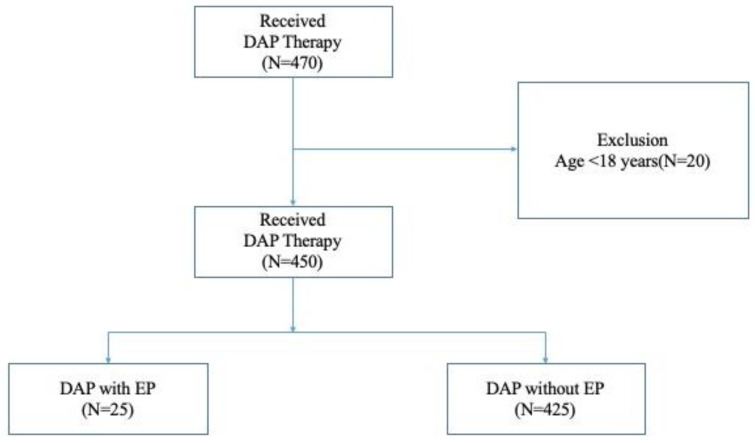
Patient selection flow chart. Abbreviations: DAP, daptomycin; EP, eosinophilic pneumonia.

**Figure 2 antibiotics-11-00254-f002:**
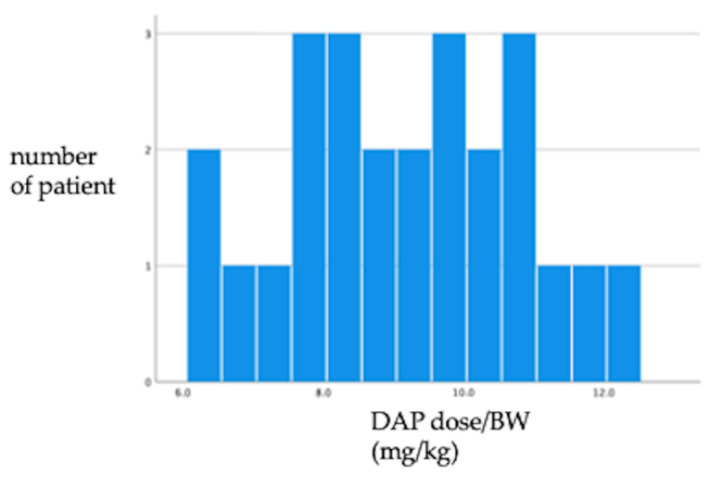
The distribution of DAP dosage /BW (mg/kg) in DAP with EP. Abbreviations: DAP, daptomycin; EP, eosinophilic pneumonia; BW, body weight.

**Table 1 antibiotics-11-00254-t001:** Characteristics of patients with DAP-induced EP.

Patient Characteristics	EP (*n* = 25)
Median duration from first DAP administration to EP onset (IQR, minimum–maximum), days	18.00 (26.5, 3–49)
Definite, *n* (%)	0 (0%)
Probable, *n* (%)	9 (36%)
Possible, *n* (%) Clinical improvement following DAP withdrawal, *n* (%)	16 (64%) 14 (56%)

Abbreviations: SD, standard deviation; DAP, daptomycin; EP, eosinophilic pneumonia, IQR: interquartile range.

**Table 2 antibiotics-11-00254-t002:** Comparison of baseline and clinical characteristics between patients with DAP-induced EP and patients without DAP-induced EP.

	DAP with EP (*n* = 25)	DAP without EP (*n* = 425)	*p*-Value
Age in years, median (IQR)	72.0 (26)	64.0 (32)	0.030
Male, *n* (%)	18 (72.0%)	249 (58.6%)	0.185
BW (IQR), kg	53.8 (17.0)	58.9 (21.6)	0.079
BMI (IQR), kg/m^2^	20.0 (6.3)	21.9 (6.0)	0.040
BW/IBW > 1, *n* (%)	9 (36.0)	192 (45.2)	0.297
BW/ABW > 1, *n* (%)	11 (44.0)	240 (56.5)	0.203
Heart failure, *n* (%)	10 (40.0)	134 (31.5)	0.378
Diabetes mellitus, *n* (%)	6 (24.0)	147 (34.6)	0.277
Respiratory disease, *n* (%)	18 (72.0)	255 (60.0)	0.233
Myocardial infarction, *n* (%)	3 (12.0)	48 (11.3)	0.914
Collagen disease, *n* (%)	2 (8.00)	41 (9.6)	0.785
Hepatic disease, *n* (%)	5 (20.0)	68 (16.0)	0.598
Solid tumor, *n* (%)	11 (44.0)	157 (36.9)	0.478
Hematological malignancy, *n* (%)	1 (4.00)	58 (13.6)	0.165
Chronic kidney disease, *n* (%)	9 (36.0)	89 (20.9)	0.076
Hemodialysis, *n* (%)	10 (40.0)	57 (13.4)	<0.001
Cerebrovascular disease, *n* (%)	2 (8.00)	66 (15.5)	0.307
Hypertension, *n* (%)	16 (64.0)	210 (49.4)	0.156
HIV, *n* (%)	0 (0)	1 (0.2)	0.808
Immunosuppressor, *n* (%)	1 (4.0)	35 (8.2)	0.448
Biological agent, *n* (%)	0 (0)	1 (0)	0.808
Steroid, *n* (%)	5 (20.0)	98 (23.1)	0.723
Statin, *n* (%)	6 (24.0)	86 (20.2)	0.650
Shock (SBP < 90), *n* (%)	7 (28.0)	158 (37.2)	0.355
qSOFA > 2, *n* (%)	11 (44.0)	193 (45.4)	0.441
DAP dosage (IQR) (mg/day)	525 (235)	350 (230)	0.046
DAP dosage /BW (IQR) (mg/kg)	9.00 (2.7)	7.50 (2.60)	<0.001
DAP dosage/IBW (IQR) (mg/kg)	8.90 (2.70)	7.40 (3.84)	0.025
DAP dosage/ABW (IQR) (mg/kg)	8.93 (2.33)	7.44 (3.25)	0.020
Total dosage of DAP (IQR) (mg)	7875 (9300)	4800 (8100)	0.032
WBC (IQR) (/μL)	8400 (2950)	7600 (5600)	0.625
Blood eosinophilia (IQR) (/μL)	419 (910)	96 (255)	<0.001
HGB (IQR) (g/dL)	10.0 (3.0)	9.50 (2.8)	0.744
PLT (IQR) (/μL)	21.2 (15.8)	21.1 (20.0)	0.643
BUN (IQR) (mg/dL)	20.0 (30.0)	18.4 (24.7)	0.968
sCr (IQR) (mg/dL)	1.01 (1.92)	0.92 (1.18)	0.929
eGFR (IQR) (mL/min/1.73 m^2^)	62.1 (78.0)	58.8 (61.9)	0.671
CCr (IQR) (mL/min/1.73 m^2^)	68.2 (66.1)	58.9 (82.4)	0.361
LDH (IQR) (IU/l)	231.5 (155)	227 (139)	0.450
CK (IQR) (IU.L)	26 (90)	47 (84)	0.102
CRP (IQR) mg/dL	10.6 (15.8)	5.39 (9.2)	0.010
Positive blood culture within one month, *n* (%)	8 (32.0)	145 (34.1)	0.828
MRSA infection, *n* (%)	4 (16.0)	29 (6.8)	0.087
CNS infection, *n* (%)	2 (8.0)	58 (13.6)	0.420
Mortality at discharge, *n* (%)	5 (20.0)	87 (20.6)	0.946
Mortality within 30 days from DAP administration, *n* (%)	4 (16.0)	59 (13.9)	
Mortality within 90 days from DAP administration, *n* (%)	4 (16.0)	76 (17.9)	
ICU admission within 30 days from admission, *n* (%)	5 (20.0)	75 (17.6)	
Mechanical intubation, *n* (%)	5 (20.0)	65 (15.3)	

MRSA or CNS infection was defined as positive culture of blood, urine, abscess, soft tissue, etc. Abbreviations: DAP, daptomycin; EP, eosinophilic pneumonia; SD, standard deviation; BW, body weight; BMI, body mass index; IBW, ideal body weight; ABW, adjusted body weight; systolic blood pressure; qsofa, quick Sequential Organ Failure Assessment score; ICU, intensive care unit; HGB, hemoglobin; PLT, platelet; BUN, blood urea nitrogen; sCr, serum creatinine; eGFR, estimated glomerular filtration rate; CCr, creatinine clearance; LDH, lactate dehydrogenase; CK, creatine kinase; CRP, C-reactive protein; MRSA, methicillin-resistant *Staphylococcus aureus*; CNS, coagulase-negative *Staphylococcus.*

**Table 3 antibiotics-11-00254-t003:** Factors associated with DAP-induced EP based on the multivariable logistic regression analysis.

	OR (95%CI)	*p*-Value
Age	1.03 (1.00–1.05)	0.042
DAP dosage/BW (mg/kg)	1.61 (1.25–2.07)	<0.001
Hemodialysis	4.42 (1.86–10.5)	0.010
MRSA infection	2.04 (0.63–6.62)	0.235

Abbreviations: DAP, daptomycin; EP, eosinophilic pneumonia; BW, body weight; BMI; ABW, adjusted body weight; MRSA, methicillin-resistant *Staphylococcus aureus*; OR, Odds Ratio; CI, Confidence Interval.

**Table 4 antibiotics-11-00254-t004:** Criteria for definite, probable, and possible DAP-induced EP (6).

**Definite** Concurrent exposure to DAP Fever Dyspnea with an increased oxygen requirement or requiring mechanical ventilation New infiltrates observed on a chest X-ray or CT scan Bronchoalveolar lavage with >25% eosinophils Clinical improvement following DAP withdrawal
**Probable** Concurrent exposure to DAP Dyspnea with increased oxygen requirement or requiring mechanical ventilation New infiltrates observed on a chest X-ray or CT scan Bronchoalveolar lavage < 25% eosinophils or peripheral eosinophilia * Clinical improvement following DAP withdrawal * Peripheral eosinophilia was defined as a peripheral blood eosinophil level above the upper limit of the normal values of the reporting laboratory or a reported elevated blood eosinophil level without any record of the actual laboratory value.
**Possible** Concurrent exposure to DAP New infiltrates observed on a chest X-ray or CT scan Clinical improvement following DAP withdrawal or patient death

Abbreviations: DAP, daptomycin; CT, computed tomography.

## Data Availability

The data presented in this study are available on request from the corresponding author (K.I.) upon reasonable request.

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
