# Peer review of "Factors Associated with Daptomycin-Induced Eosinophilic Pneumonia"

_antibiotics, 2022, doi:10.3390/antibiotics11020254_

Round 1

Reviewer 1 Report

In the present manuscript, the authors have done a retrospective observational study of risk factors for DAP-induced eosinophilic pneumonia. It is a well-designed study and the concepts of daptomycin eosinophilic pneumonia: definite, probable and possible are very clear. Some recent work has been published about the risk factors of DAP-induced eosinophilic pneumonia. Despite this, there are some details to highlight in this work. The details on daptomycin dosage with the concepts of IBW, ABW are a good appreciation. Some other works found comorbidities as a general risk factor and in this work the haemodialysis has been the most important.

I think is a very good work, but there is not too much new information. On the other hand, it is true that this retrospective study can be useful to strengthen concepts already discussed in previous works.

Specific minor comments:

- line 40 Discussion. “should be increased from every 24 h to every 48 h” I think the correct sentence is “should be decreased from every 24 h to every 48 h”

-In my opinion table 1 is not necessary. The information is repeated in the manuscript.

- it is recommended that you use the same number of decimal places in the tables (p value).

Author Response

- line 40 Discussion. “should be increased from every 24 h to every 48 h” I think the correct sentence is “should be decreased from every 24 h to every 48 h”

Kazuhiro Ishikawa> Thank you for pointing that out. I've changed “increased” to "decreased" in page 6 line 160.

-In my opinion table 1 is not necessary. The information is repeated in the manuscript.

Kazuhiro Ishikawa> Thank you for pointing this out. Table 1 is important for this study, so we have revised the sentence as follows

In page 2 line 64-65,

The median duration from the first DAP dose administration to EP onset among the 25 patients with EP was 18.00 (interquartile range [IQR], 26.5; minimum-maximum, 349) days. The number of patients with definite, probable, and possible EP is as follows(Table 1). No patients underwent BAL, so the number of patients with definite EP was zero. Fourteen patients with possible EP improved clinically after discontinuation of DAP, but two patients died of heart failure.

- it is recommended that you use the same number of decimal places in the tables (p value).

Kazuhiro Ishikawa> Thank you for pointing this out. I have aligned the numbers after the decimal point in the tables (p value).

Reviewer 2 Report

Ishikawa, K and co-authors investigated the risk factors associated with eosinophilic pneumonia induced by daptomycin in their retrospective observational study. Although patients who met the criteria for definite EP were not included in the presented study, this examination had important message that the elder and a hemodialysis patient are particularly should be monitored carefully when administering DAP for treatment of MRSA infection.

I would like to accept this manuscript after revised several points.

Although author showed patients received more than 9 mg/kg of DAP more likely to cause EP, we could not figure out how many patients received DAP over 9 mg/kg in this study. Please make a figure showing dose distribution for DAP with EP. For example, indicate the groups 4 mg (4-6 mg), 6 mg (6-8 mg), 8mg (8-10 mg), 10mg (over 10mg) of DAP dose.

Majority patients in this study received DAP administration for what kind of infection or purpose? Table 2 shows that MRSA infection is included only 16% and 6.8% between patients DAP-induced EP and patients without DAP-induced EP, respectively. Please provide a detailed description for the reason why eligible patients received DAP therapy.

The ratio of complicating EP related with DAP was relatively high (5.5%; 25 of 540 patients) compared with previous reports in the present study. Please explain why the ratio of complicating EP was comparatively high in this study?

Please describe, in author’s opinion, why old age was dominant risk factors for DAP induced EP? Previous studies also showed the risk factors for DAP induced EP was age over 70.

Author Response

-Although author showed patients received more than 9 mg/kg of DAP more likely to cause EP, we could not figure out how many patients received DAP over 9 mg/kg in this study. Please make a figure showing dose distribution for DAP with EP. For example, indicate the groups 4 mg (4-6 mg), 6 mg (6-8 mg), 8mg (8-10 mg), 10mg (over 10mg) of DAP dose.

 Kazuhiro Ishikawa> Thank you for the suggestion. I have added sentence in page 3 line 89-90 and figure 2 regarding the distribution of DAP/BW in DAP with EP.

-Majority patients in this study received DAP administration for what kind of infection or purpose? Table 2 shows that MRSA infection is included only 16% and 6.8% between patients DAP-induced EP and patients without DAP-induced EP, respectively. Please provide a detailed description for the reason why eligible patients received DAP therapy.

Kazuhiro Ishikawa> Thank you for pointing this out. In this study, we were not able to extract the diagnosis of the patients who received daptomycin. MRSA infections are those in which MRSA could be identified by blood, urine, abscess, soft tissue, and so on in page 3 Line 89.  Therefore, the definition has been added in the result and table 2.

-The ratio of complicating EP related with DAP was relatively high (5.5%; 25 of 540 patients) compared with previous reports in the present study. Please explain why the ratio of complicating EP was comparatively high in this study?

Kazuhiro Ishikawa> Thank you for pointing this out. The EP rate in the previous study, which is also cited in the text, was 4.8%. The rate in EP of this study is slightly higher. We could assume that it is because there were fewer patients with renal dysfunction and no hemodialysis patient which are risk factor for EP in the previous study, but many these patients were included in this study.

Antibiotics (Basel). 2021 Apr 16;10(4):446. doi: 10.3390/antibiotics10040446.

Please describe, in author’s opinion, why old age was dominant risk factors for DAP induced EP? Previous studies also showed the risk factors for DAP induced EP was age over 70.

Kazuhiro Ishikawa> Thank you for pointing this out. Both the present study and the previous study that we cited had the same result that DAP with EP is more common in the elderly. Neither this study nor the previous study found any reason why the elderly are at risk for EP.

I supposed that bioaccumulation of daptomycin occurs more in the elderly than in the young due to reduced organ function.

.Antibiotics (Basel). 2021 Apr 16;10(4):446. doi: 10.3390/antibiotics10040446.

Reviewer 3 Report

  • The bibliographic references need to be numbered since in the text they refer to citations with a specific number.
  •  
  • Lack of definition of eosinophilic pneumonia in the introduction.
  •  
  • What diagnosis did the patients have when the treatment with Daptomycin was prescribed? Were they soft tissue infections, bacteremia, etc? endocarditis? 
  •  
  • An Index would be missing to define the severity of the sample of patients analyzed, such as: Charlson Index, McCabe Index.
  •  
  • The high number of eosinophilic pneumonias described by the authors is striking. Some imaging test of these pneumonia would be missing.

Author Response

The bibliographic references need to be numbered since in the text they refer to citations with a specific number.

Kazuhiro Ishikawa> Thank you for pointing this out. The number in reference was missing, so I fixed it.

Lack of definition of eosinophilic pneumonia in the introduction.

Kazuhiro Ishikawa> Thank you for pointing this out, I have added the definition to the Introduction on page1 line 45-46.

EP is characterized by marked accumulations of infiltrating eosinophils in the alveolar space and the interstitium. 

What diagnosis did the patients have when the treatment with Daptomycin was prescribed? Were they soft tissue infections, bacteremia, etc? endocarditis? 

Kazuhiro Ishikawa> Thank you for pointing this out. Because we focused only on the side effects of daptomycin in this study, we were able to extract the culture results which can be identified pathogen, but not the diagnosis. 

An Index would be missing to define the severity of the sample of patients analyzed, such as: Charlson Index, McCabe Index.

Kazuhiro Ishikawa> Thank you for pointing this out. We did not estimate the Charson index in this study, but we show that there was no significant difference in compared with underlying disease, qSofa or shock between DAP with EP and DAP without EP.

The high number of eosinophilic pneumonias described by the authors is striking. Some imaging test of these pneumonia would be missing.

Kazuhiro Ishikawa> Thank you for pointing this out. The EP rate in the previous study, which is also cited in the text, was 4.8%. The rate in EP  of this study is slightly higher. We could assume that it is because there were fewer patients with renal dysfunction and no hemodialysis patient which are risk factor for EP in the previous study, but many of these patients were included in this study.

Antibiotics (Basel). 2021 Apr 16;10(4):446. doi: 10.3390/antibiotics10040446.